# What Makes Bariatric Operations Difficult–Results of a National Survey

**DOI:** 10.3390/medicina55060218

**Published:** 2019-05-28

**Authors:** Piotr Major, Tomasz Stefura, Maciej Walędziak, Michał Janik, Michał Pędziwiatr, Michał Wysocki, Mateusz Rubinkiewicz, Jan Witowski, Jacek Szeliga, Andrzej Budzyński

**Affiliations:** 12nd Department of General Surgery, Jagiellonian University Medical College, 31-501 Krakow, Poland; piotr.major@uj.edu.pl (P.M.); michal.pedziwiatr@uj.edu.pl (M.P.); michal92wysocki@gmail.com (M.W.); mrubinkiewicz@gmail.com (M.R.); andrzej.budzynski@uj.edu.pl (A.B.); 2Centre for Research, Training and Innovation in Surgery (CERTAIN Surgery), 31-501 Krakow, Poland; jwitos@gmail.com; 3Students’ Scientific Group at 2nd Department of General Surgery, Jagiellonian University Medical College, 31-501 Krakow, Poland; 4Department of General, Oncological, Metabolic and Thoracic Surgery, Military Institute of Medicine, 04-141 Warsaw, Poland; maciej.waledziak@gmail.com (M.W.); janiken@gmail.com (M.J.); 5Department of General, Gastroenterological, and Oncological Surgery Collegium Medicum, Nicolaus Copernicus University, 87-100 Torun, Poland; jacky2@wp.pl

**Keywords:** bariatric surgery, sleeve gastrectomy, Roux-en-Y gastric bypass, one-anastomosis gastric bypass, technical difficulties

## Abstract

*Background and objective:* The most commonly performed bariatric procedures include laparoscopic sleeve gastrectomy (LSG), laparoscopic Roux-en-Y gastric bypass (LRYGB), and one anastomosis gastric bypass-mini gastric bypass (OAGB-MGB). A study comparing the degree of difficulty among those procedures could serve as a guide for decision making in bariatric surgery and further improve training programs for general surgery trainees. The aim of this study was to compare the subjective level of technical difficulty of LSG, LRYGB, and OAGB–MGB as perceived by surgeons and surgical residents. *Materials and Methods*: An anonymous internet-based survey was designed to evaluate the subjective opinions of surgeons and surgical residents in training in Poland. It covered baseline characteristics of the participants, difficulty of LSG, OAGB-MGB, LRYGB and particular stages of each operation assessed on a 1–5 scale. *Results*: Overall, 70 surgeons and residents participated in our survey. The mean difficulty degree of LSG was 2.34 ± 0.89. The reinforcing staple line with sutures was considered most difficult stage of this operation (3.17 ± 1.19). The LRYGB operation had an average difficulty level of 3.87 ± 1.04. Creation of the gastrojejunostomy was considered the most difficult stage of LRYGB with a mean difficulty level (3.68 ± 1.16). Responders to our survey assessed the mean degree of difficulty of OAGB-MGB as 2.34 ± 0.97. According to participating surgeons, creating the gastrojejunostomy is the most difficult phase of this operation (3.68 ± 1.16). *Conclusion:* The LSG is perceived by surgeons as a relatively easy operation. The LRYGB was considered to be the most technically challenging procedure in our survey. Operative stages, which require intra-abdominal suturing with laparoscopic instruments, seem to be the most difficult phases of each operation.

## 1. Introduction

Currently there are multiple bariatric operations available, which achieve both satisfactory weight loss and the remission or resolution of obesity-related comorbidities [1]. The most commonly performed bariatric procedures include laparoscopic sleeve gastrectomy (LSG), laparoscopic Roux-en-Y gastric bypass (LRYGB), and mini-gastric bypass, also known as one anastomosis gastric bypass (OAGB-MGB) [2,3]. Current guidelines for the surgical treatment of obesity do not have precise recommendations for choosing a suitable bariatric operation [4,5,6,7]. The decision to choose the most suitable procedure is a complex process and should not be based only on a patient’s preferences [8]. The optimal solution is often reached by a multidisciplinary team that is able to comprehensively assess the risk-benefit ratio of each procedure and identify essential bariatric outcome targets for a specific patient [9]. A surgeon’s operative experience and technical proficiency is also one of the most important factors in the decision making process [10,11,12]. However, very frequently, subjective criteria related to how surgeons perceive a particular procedure are most important. Obviously, with the absence of strict and clear guidelines, surgeons tend to choose the operation that is, in their opinion, easier to perform, more appealing, and more effective. On the other hand, it is known that operative technique and choice of the procedure are significant predictors of potential readmission after bariatric surgery [13].

We were not able to find a study comparing the degree of difficulty of common bariatric procedures, comparable to survey of Jamali, which evaluated this issue in respect to colorectal surgery [14]. A similar study could serve as a guide for decision making in bariatric surgery. Moreover, a survey in the group of medical doctors at different stages of surgical training would provide an original insight into characteristics of previously mentioned operations and help to further improve training programs and curriculums for general surgery trainees.

Our aim was to compare the subjective level of technical difficulty of LSG, LRYGB, and OAGB–MGB as perceived by surgeons and surgical residents. We also investigated the technical difficulties associated with particular stages of each procedure and verified several technical aspects of previously mentioned operations.

## 2. Material and Methods

### 2.1. Study Design

An anonymous internet-based survey was designed to evaluate the subjective opinions of surgeons and surgical residents in training in Poland. The survey was distributed between January and June 2018 using mailing lists and the official website of the Metabolic and Bariatric Surgery Chapter of the Polish Surgical Society. The questionnaire was supplemented with comprehensive instructions on how to complete the survey.

### 2.2. Inclusion and Exclusion Criteria

We included general surgeons and medical doctors in general surgery residency programs who are actively working in departments performing surgical treatment of obesity. We excluded physicians with specializations other than surgery, non-clinical healthcare professionals, interns, and students. We also excluded participants from countries other than Poland.

### 2.3. Survey

The questionnaire was divided into five parts, which included questions concerning:Baseline characteristics of the participant: age, sex, stage of surgical training-resident/specialist, experience in general surgery (years), experience in bariatric surgery (years), and number of performed LSGs, LRYGBs, and OAGB-MGBs.The incidence of technical difficulties during bariatric operations in groups of patients based on the Body Mass Index (BMI) 35–40, 41–50, 51–60, and >60 (five-point Likert scale).Difficulty of the LSG and particular stages of this operation (five-point Likert scale) including: creation of pneumoperitoneum, visualization of the operative field, releasing the adhesions, liver retraction, dissection off the greater curvature of the stomach from the gastro-colic ligament, dissection of the short gastric vessels, calibration of the sleeve with the probe, resection of the stomach with stapler, staple line reinforcement with stitches, control of the potential hemorrhage from the staple line, leak test, removal of the resected portion of the stomach from the peritoneal cavity, and suturing the port sites. We also asked our respondents about their method for stopping bleeding from the staple line during the LSG.Difficulty of the LRYGB and particular stages of this operation (five-point Likert scale) including: creation of pneumoperitoneum, visualization of the operative field, releasing the adhesions, liver retraction, dissection of the fundus of the stomach, creation of the pouch, division of the jejunum into the alimentary and the enzymatic limb, creation of the gastrojejunostomy, dissection of the greater omentum, measuring the length of jejunum to create appropriate jejunojejunostomy, creation of the jejunojejunostomy, closure of the Petersen space and the intermesenteric space, and suturing the port sites. We also asked our respondents about their method for creating the gastro-jejunal anastomosis during the LRYGB.Difficulty of the OAGB-MGB and particular stages of this operation (five-point Likert scale) including: creation of pneumoperitoneum, visualization of the operative field, releasing the adhesions, liver retraction, dissection of the fundus of the stomach, creation of the pouch, measuring the length of jejunum to create appropriate gastrojejunostomy, creation of the gastrojejunostomy, closure of the Petersen space and the intermesenteric space, and suturing the port sites.

To assess difficulty, a five point Likert-like scale was used, where 1 corresponded to “Very low difficulty degree”, 2 corresponded to “Low difficulty degree”, 3 corresponded to “Medium difficulty degree”, 4 corresponded to “High difficulty degree” and 5 corresponded to “Very high difficulty degree”.

### 2.4. Statistical Analysis

All data were analyzed using Statistica version 13.1PL (StatSoft Inc., Tulsa, OK, USA). The normal distribution was checked using a Shapiro-Wilk test. The results are presented as a mean with standard deviation (SD) or median with interquartile range (IQR). A comparison of the baseline patients’ data was made using Student’s *t*-test or Mann-Whitney’s test. Pearson’s test was used to verify the correlation between the degree of difficulty and the experience in working as a general surgeon reported in years. Results were considered statistically significant at *p* < 0.05.

### 2.5. Ethics Approval and Consent to Participate in the Study

Informed consent was obtained from all individual participants included in the study. The data were completely anonymized. The study protocol was approved by the board of the Metabolic and Bariatric Surgery Chapter of the Polish Surgical Society, and the study was conducted under its supervision. All procedures have been performed in accordance with the ethical standards laid down in the 1964 Declaration of Helsinki and its later amendments (Fortaleza).

## 3. Results

Overall, 70 surgeons and residents from 16 Polish surgical centers participated in our survey. The group of respondents included 51 surgeons (72.86%) and 19 surgical residents in training (27.14%). The study group included 57 males (81.43%) and 13 females (18.57%). The mean age was 41.04 ± 11.18 years. The average experience working as a general surgeon was 15.31 ± 11.78 years. The participants’ experience in the surgical treatment of obesity was 7.39 ± 5.91 years on average, and they performed a median of 75 (20–200) LSGs, 10 (50) LRYGBs, and 2 (0–5) OAGB-MGBs as a first operator (among surgeons performing particular procedure). Study group characteristics are presented in Table 1.

Surgeons reported the highest incidence of technical difficulties during bariatric operation in patients with BMI above 60. The incidence of difficulties seems to correlate with increasing BMI (Figure 1).

The mean difficulty degree of LSG was scored as 2.34 ± 0.89 points. It did not differ between residents and certified surgeons (2.56 ± 0.62 vs. 2.33 ± 1.09, *p* = 0.42). There was no correlation between reported difficulty degree of LSG with years of experience in general surgery (r = ‒0.01, *p* = 0.94). The reinforcing staple line with sutures was considered the most difficult stage of this operation (3.17 ± 1.19), followed by dissection of short gastric vessels (2.71 ± 0.99), resection of the stomach with stapler (2.67 ± 1.16), and control of the potential hemorrhage from the staple line (2.64 ± 1.11). The reported difficulty degree of staple line reinforcement with sutures and control of the potential hemorrhage from the staple line was higher among residents than surgeons (*p* = 0.02 and *p* = 0.03, respectively). There was a negative correlation between the reported difficulty degree of staple line reinforcement with sutures with years of experience in general surgery (r = −0.27, *p* = 0.02) (Table 2). Surgeons most commonly reported clips as the preferred method of bleeding control (49–70%), followed by manual suturing (9–12.9%), coagulation (7–10%), and hemostatic material (1–1.4%).

LRYGB had an average difficulty level of 3.87 ± 1.04 points. Residents reported a significantly higher degree of difficulty than surgeons (4.47 vs. 3.74, *p* = 0.03). There was a significant correlation between the reported difficulty degree of LRYGB and years of experience in general surgery (r = −0.275, *p* = 0.03). The creation of the gastrojejunostomy was considered the most difficult stage of LRYGB, with a mean difficulty level of 3.68 ± 1.16, followed by creation of the jejunojejunostomy (3.6 ± 1.02), dividing the jejunum into the alimentary and the enzymatic limbs (3.26 ± 1.1), closure of the Petersen space and the inter-mesenteric space (3.18 ± 1.1), and creation of the pouch (3.15 ± 1.03). The reported difficulty degree of division of the jejunum into the alimentary and the enzymatic limbs and creation of the gastrojejunostomy was higher among residents than surgeons (*p* < 0.01 and *p* = 0.04, respectively). There was a negative correlation between the reported difficulty degree of division of the jejunum into the alimentary and the enzymatic limbs with years of experience in general surgery (r = –0.31, *p* = 0.01) (Table 3). The majority of participants in our study reported use of the circular stapler as the primary method of creating the gastrojejunostomy (33–47.1%). Furthermore, 24 respondents (34.3%) reported a linear stapler, and one surgeon reported manual suturing of the anastomosis (1.4%).

Responders to our survey assessed the mean degree of difficulty of OAGB-MGBs as 2.34 ± 0.97 points, which was significantly higher among residents in comparison with certified surgeons (4 ± 0.73 vs. 3.22 ± 0.98, *p* < 0.01). There was a significant correlation between reported difficulty degree of OAGB-MGB and years of experience in general surgery (r = −0.424, *p* < 0.01). According to participating surgeons, creating the gastrojejunostomy is the most difficult phase of this operation (3.68 ± 1.16). The next most challenging stages were: measuring the length of jejunum to creation of appropriate gastrojejunostomy (3.26 ± 1.1), closure of the Petersen space and the inter-mesenteric space, (3.18 ± 1.1), and creation the pouch (3.15 ± 1.03). The reported difficulty degree of creating the pouch, measuring the length of jejunum to create an appropriate gastrojejunostomy, and creating the gastrojejunostomy, was higher among residents than surgeons (*p* = 0.04, *p* = 0.02, and *p* < 0.01, respectively). There was a negative correlation between the reported difficulty degree of dissecting the fundus of the stomach, creating the pouch, measuring the length of jejunum to create appropriate gastrojejunostomy, and creating the gastrojejunostomy with years of experience in general surgery(r = −0.32, *p* = 0.04 r = −0.41, *p* < 0.01, r = −0.41, *p* < 0.01, and r = −0.49, *p* < 0.01, respectively) (Table 4).

## 4. Discussion

Our survey-based study aimed to investigate difficulty of the most commonly performed bariatric operations: LSG, LRYGB, and OAGB–MGB, and identify which stages of each procedure seem to be the most technically challenging for a surgeon. In the subjective opinion of participants, LRYGB was the most difficult. We also observed that those stages of the investigated procedures that require manual intra-abdominal suturing under laparoscopic vision are particularly demanding for operating surgeons.

Currently, there is no universal grading system to assess the technical difficulty of an operation. According to Iwashita et al., in the case of laparoscopic cholecystectomy, objective intra-operative findings (factors related to inflammation of the gallbladder or additional findings of the gallbladder and its surroundings) should be considered indices of technical difficulty, instead of operative time or conversion rate [15]. However, this approach does not allow one to compare the difficulty of different operations, even in uneventful cases. We aimed to assess the technical challenges associated with each operation by conducting a survey among surgeons dealing with the surgical treatment of obesity on a daily basis by obtaining their subjective opinions.

Technical obstacles often negatively impact surgical outcomes. For instance, colorectal operations, which are considered difficult, are often associated with inferior postoperative oncological outcomes [16,17]. Technical problems also increase the incidence of complications after various bariatric procedures [18]. Identifying types of operations and their stages with the highest level of difficulty may help to improve outcomes after the surgical treatment of obesity by pointing out elements that require further improvement and more emphasis while designing training programs.

Both underweight and obese patients achieved worse outcomes after surgery. Obesity significantly increases operative time and mortality rates [19]. Postoperative morbidity is also significantly higher among morbidly obese individuals. This effect is particularly pronounced in open abdominal surgery; therefore, modern bariatric procedures are usually performed laparoscopically [20,21]. Generally high body mass significantly increases the difficulty of surgical operation and post-operative care [22]. According to the results obtained by our survey, surgeons recognize increasing BMI as a factor that correlates with the incidence of technical difficulties during bariatric operations.

Participants reported the LRYGB to be the most technically difficult procedure, followed by the OAGB–MGB, and the LSG, which were perceived as operations with a comparable, low difficulty degree. Nevertheless, the operative techniques for OAGB-MGB and LRYGB have substantial similarities [12]. Therefore, we believe that surgeons at the beginning of their experience should master their skills with the LSG and later progress to OAGB-MGB and LRYGB. This is consistent with previous studies that indicate the LSG to be a less technically challenging operation with a shorter operative time than the LRYGB [23,24]. The operative time of the OAGB-MGB is slightly longer, but comparable with the LSG, and remains significantly shorter than the operative time of LRYGB. Therefore, this procedure could serve as an intermediate step between LSG and LRYGB during the training of a bariatric surgeon [25,26].

The common steps of all three procedures, which are also present in various other laparoscopic operations, such as the creation of pneumoperitoneum, visualization of the operative field, releasing the adhesions, liver retraction, and suturing the port sites, were all associated with a comparable, low degree of technical difficulty. The most challenging stages of operations included in this study were those which required manual suturing with laparoscopic instruments, such as reinforcing the staple line during the LSG or closing the stapler insertion places during the LRYGB. Teaching novice surgeons the skill of laparoscopic suturing remains difficult and requires long and specialized training [27]. An intensive training course may shorten the learning curve [28]. It seems, however, that proficiency is often associated with a high aptitude of the surgical trainee. There is a group of candidates who will not be able to excel in this area, despite the time-consuming training program [29].

Staple line bleeding is a common intra-operative adverse event during LSG. Choosing the appropriate length of staples is believed to play some role in the prevention of this complication [30]. A study by Chakravartty et al. reports that tranexamic acid could be an additional, simple, and cost-effective method for prophylaxis of staple line bleeding [31]. Possible methods for managing the bleeding during an operation include compression, ligation, clipping, and suturing [32]. Participants in the presented study most commonly reported using surgical clips for stopping bleeding from the staple line during the LSG. It is, however, speculated that this procedure may interfere with revisional operations, which are reported to be more and more frequent.

Past comparisons of hand-sewn, linear-stapled, and circular-stapled gastrojejunostomies during LRYGB usually did not reveal considerable differences in outcomes [33,34]. Several previous studies reported a significantly lower incidence of infectious complications after linear-stapled gastrojejunostomies in comparison with circular-stapled ones, which could be related to the passage of a contaminated circular stapler through the abdominal wall [35,36,37]. Nevertheless, according to our results, the most popular method of performing gastrojejunostomy during LRYGB was the use of a circular stapler.

Our study has several limitations. The group of 70 participants is relatively low, and all responding surgeons currently work in Polish bariatric centers. Thus, the results should not be translated beyond the study group examined. Our survey did not include a precise comparison of the operative technique and selected only specific stages of each procedure. Nevertheless, we believe that potential bias resulting from missed discrepancies in the operative technique was minimized. Moreover, we did not include other, less popular bariatric procedures, such as adjustable gastric banding, biliopancreatic diversion with duodenal switch, single anastomosis duodeno-ileal bypass, etc. Having said that, those procedures are rarely performed, and the obtained data may be of limited value. The quality of this study could be improved with the participation of a larger group of surgeons, preferably from many countries. It is also important to re-evaluate bariatric operations after a few years, as trends in bariatric surgery change.

## 5. Conclusions

The LSG is perceived by surgeons to be a relatively easy operation to perform and start training with, while LRYGB was considered to be the most technically challenging. Operative stages that require intra-abdominal suturing with laparoscopic instruments seem to be the most difficult phases in all studied procedures. Thus, particular emphasis should be placed on this aspect of training in bariatric surgery. Clipping is the most popular method for bleeding control from the staple line, and circular-stapler remains the most widely used technique for creating the gastrojejunostomy.

## Figures and Tables

**Figure 1 medicina-55-00218-f001:**
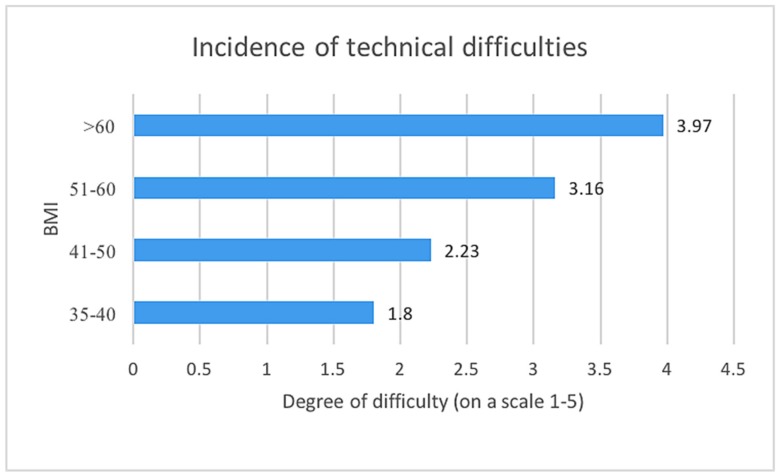
Incidence of technical difficulties during bariatric operations, group by patients body mass index (BMI).

**Table 1 medicina-55-00218-t001:** Study group characteristics. Laparoscopic sleeve gastrectomy (LSG); laparoscopic Roux-en-Y gastric bypass (LRYGB); one anastomosis gastric bypass-mini gastric bypass (OAGB-MGB); interquartile range (IQR); standard deviation (SD).

Variable	Overall *n* = 70 (100%)	Residents *n* = 19 (27.14%)	Surgeons *n* = 51 (72.86%)	*p*
Males, *n* (%)	57 (81.43)	10 (52.63)	47 (92.16)	<0.01
Females, *n* (%)	13 (18.57)	9 (47.37)	4 (7.84)	
Age, years, mean ± SD	41.04 ± 11.10	30.28 ± 2.37	44.92 ± 10.54	<0.01
Experience in general surgery, years, mean ± SD	15.31 ± 11.78	3.92 ± 2.43	19.64 ± 10.99	<0.01
Experience in bariatric surgery, years, mean ± SD	7.39 ± 5.91	2.82 ± 1.98	9.09 ± 5.99	<0.01
Number of performed LSGs, median (IQR)	75 (20–200)	5 (0–40)	130 (40–230)	<0.01
Number of performed LRYGBs, median (IQR)	10 (0–50)	0 (0–0)	25 (2–74)	<0.01
Number of performed OAGB-MGBs, median (IQR)	2 (0–5)	0 (0–0)	5 (0–8)	<0.01

**Table 2 medicina-55-00218-t002:** Difficulty degree of each stage of the LSG.

Stage of the Operation	Overall Degree of Difficulty on a Scale 1–5 (Mean ± SD)	Degree of Difficulty Among Residents (Mean ± SD)	Degree of Difficulty Among Surgeons (Mean ± SD)	*p*	Correlation Between Degree of Difficulty and Years of Experience in General Surgery	
					r	*p*
Creation of pneumoperitoneum	1.68 ± 0.78	1.61± 0.78	1.71 ± 0.78	0.66	0.11	0.37
Visualization of the operative field	1.77 ± 0.81	1.78 ± 0.73	1.76 ± 0.84	0.95	−0.06	0.63
Release of adhesions	2.18 ± 0.96	2.39 ± 0.98	2.1 ± 0.95	0.28	−0.06	0.63
Liver retraction	1.72 ± 0.83	1.82 ± 0.64	1.69 ± 0.88	0.56	0.04	0.75
Dissection of the greater curvature of the stomach from the gastro-colic ligament	2.14 ± 0.99	2.39 ± 0.78	2.01 ± 1.04	0.23	0.03	0.79
Dissection of the short gastric vessels	2.71 ± 0.99	3.06 ± 0.73	2.59 ± 1.04	0.08	−0.19	0.13
Calibration with the probe	2.13 ± 0.97	2.39 ± 0.98	2.04 ± 0.96	0.19	−0.17	0.18
Resection of the stomach with a stapler	2.67 ± 1.16	3 ± 1.03	2.55 ± 1.19	0.16	−0.08	0.53
Staple line reinforcement with sutures	3.17 ± 1.19	3.77 ±1.03	2.96 ± 1.18	0.02	−0.27	0.03
Control of the potential hemorrhage from the staple line	2.64 ± 1.11	3.11 ± 1.023	2.47 ± 1.1	0.03	−0.16	0.18
Leak test	1.61 ± 0.79	1.44 ± 0.61	1.67 ± 0.84	0.31	0.16	0.20
Removal of the resected portion of the stomach from the peritoneal cavity	1.71 ± 0.73	1.78 ± 0.88	1.69 ± 0.68	0.65	0.03	0.80
Suturing the port sites	1.63 ± 0.86	1.33 ± 0.49	1.74 ± 0.94	0.09	0.05	0.67

**Table 3 medicina-55-00218-t003:** Difficulty degree of each stage of the LRYGB.

Stage of the Operation	Overall Degree of Difficulty on a Scale 1−5 (Mean ± SD)	Degree of Difficulty Among Residents (Mean ± SD)	Degree of Difficulty Among Surgeons (Mean ± SD)	*p*	Correlation Between Degree of Difficulty and Years of Experience in General Surgery	
					r	*p*
Creation of pneumoperitoneum	1.69 ± 0.82	1.65 ± 0.86	1.71 ± 0.82	0.79	0.08	0.53
Visualization of the operative field	1.88 ± 0.83	2.12 ± 0.93	1,8 ± 0.79	0.18	−0.18	0.17
Release of adhesions	2.33 ± 0.93	2.41 ± 1	2.3 ± 0.9	0.66	0.02	0.91
Liver retraction	1.85 ± 0.79	2.06 ± 0.75	1.78 ± 2.01	0.21	0.02	0.91
Dissection of the fundus of the stomach	2.56 ± 1.03	2.89 ± 0.93	2.43 ± 1.04	0.12	−0.11	0.41
Creation of the pouch	3.15 ± 1.03	3.38 ± 0.81	3.07 ± 1.1	0.31	−0.13	0.34
Division of the jejunum into the alimentary and the enzymatic limbs	3.26 ± 1.1	3.88 ± 0.86	3.02 ± 1.1	<0.01	−0.31	0.01
Creation of the gastrojejunostomy	3.68 ± 1.16	4.18 ± 0.88	3.49 ± 1.2	0.04	−0.12	0.37
Dissection of the greater omentum	2.27 ± 0.88	2.59 ± 0.87	2.14 ± 0.86	0.08	-0.06	0.67
Measuring the length of jejunum to create appropriate jejunojejunostomy	2.87 ± 1.02	3.29 ± 1.11	2.71 ± 1.06	0.06	−0.22	0.08
Creation of the jejunojejunostomy	3.6 ± 1.02	4 ± 0.79	3.4 ± 1.06	0.05	−0.11	0.39
Closure of the Petersen space and the intermesenteric space	3.18 ± 1.1	3.23 ± 0.97	3.16 ± 1.15	0.82	0.06	0.67
Suturing the port sites	1.7 ± 0.86	1.41 ± 0.51	1.82 ± 0.94	0.09	0.01	0.96

**Table 4 medicina-55-00218-t004:** Difficulty degree of each stage of the OAGB-MGB.

Stage of the Operation	Overall Degree of Difficulty on a Scale 1–5 (Mean ± SD)	Degree of Difficulty Among Residents (Mean ± SD)	Degree of Difficulty Among Surgeons (Mean ± SD)	*p*	Correlation Between Degree of Difficulty and Years of Experience in General Surgery	
					r	*p*
Creation of pneumoperitoneum	1.69 ± 0.82	1.39 ± 0.51	1.77 ± 0.82	0.13	0.09	0.56
Visualization of the operative field	1.87 ± 0.83	1.83 ± 0.84	1.88 ± 0.94	0.92	−0.15	0.34
Release of adhesions	2.33 ± 0.9	2.39 ± 0.87	2.31 ± 0.93	0.81	−0.17	0.29
Liver retraction	1.85 ± 0.79	2.08 ± 0.76	1.77 ± 0.82	0.25	−0.25	0.11
Dissection of the fundus of the stomach	2.56 ± 1.03	2.77 ± 0.73	2.37 ± 0.89	0.16	−0.32	0.04
Creation of the pouch	3.15 ± 1.03	3.31 ± 0.75	2.63 ± 1.03	0.04	−0.41	<0.01
Measuring the length of jejunum to create appropriate gastrojejunostomy	3.26 ± 1.1	3.54 ± 1.05	2.6 ± 1.16	0.02	−0.41	<0.01
Creation of the gastrojejunostomy	3.68 ± 1.16	4.08 ± 0.76	3.1 ± 1.13	<0.01	−0.49	<0.01
Closure of the Petersen space and the intermesenteric space	3.18 ± 1.1	3.15 ± 0.9	2.74 ± 1.06	0.23	0.27	0.10
Suturing the port sites	1.7 ± 0.86	1.69 ± 0.95	1.83 ± 0.91	0.65	−0.04	0.78

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
