# Peer review of "What Makes Bariatric Operations Difficult–Results of a National Survey"

_medicina, 2019, doi:10.3390/medicina55060218_

Reviewer 1 Report

Authors have made all required changes based on the comments. No other changes needed from my standpoint.

Reviewer 2 Report

The authors responded sufficiently to the comments raised by the reviewers and the Editor.

This manuscript is a resubmission of an earlier submission. The following is a list of the peer review reports and author responses from that submission.

Round  1

Reviewer 1 Report

This is a straightforward study and the manuscript is well written. The results are presented clearly and the discussion is comprehensive.

The results of this survey to some degree are expected and underline the critical points of advanced laparoscopic surgery. Although there are sufficient data on the responders' status, I would like to see an analysis of possible associations between degree of difficulty experienced and surgeon’s experience on bariatric surgery (number of operations performed) or residents versus qualified surgeons.

Another minor point: Please make clear that results are presented as mean ± SD in the manuscript and in tables.

Reviewer 2 Report

This is a very interesting study on the level of difficulty of bariatric procedures. My questions and suggestion:

- please clarify the level of residents involved in this study and how much exposure they have had with these procedures. 

- Is it possible to do a subgroup anaylsis for "surgeons" and "residents"

- please clarify if the Likert scale was (1, 2, 3, 4, 5) or something like "very difficult, difficult, ...". You may verify with an statistician about how to analyze the results. They may advise to do chi square after categorizing changing the variables to "difficult vs not difficult". 

- Also please verify if you are better to report values of difficulty degree as mean (SD) or median (IQR), based on being in normal distribution or not. 

- You need to explain the values in the tables (e.g, mean±SD)